# Gd/Y Hydroxide Nanosheets as Highly Efficient T_1_/T_2_ MRI Contrast Agents

**DOI:** 10.3390/nano11010017

**Published:** 2020-12-24

**Authors:** Xin Li, Zhenhai Xue, Jinfeng Xia, Guohong Zhou, Danyu Jiang, Mengting Dai, Wenhui Wang, Jiayan Miu, Yuerong Heng, Cuiyan Yu, Qiang Li

**Affiliations:** 1School of Chemistry and Molecular Engineering, East China Normal University, Shanghai 200062, China; xinl728@stu.ecnu.edu.cn (X.L.); 52194300002@stu.ecnu.edu.cn (M.D.); wwh1017111@163.com (W.W.); m13750810742@163.com (J.M.); h18783906501@163.com (Y.H.); yucuiyan012@163.com (C.Y.); 2Shanghai Institute of Ceramics, Chinese Academy of Sciences, 1295 Dingxi Road, Shanghai 200050, China; zhxue@mail.sic.ac.cn (Z.X.); xiajf@mail.sic.ac.cn (J.X.); sic_zhough@mail.sic.ac.cn (G.Z.); dyjiang@mail.sic.ac.cn (D.J.)

**Keywords:** Gd-doped nanosheets, dual-modal contrast agent, MRI, PAA

## Abstract

To develop highly efficient T_1_/T_2_ magnetic resonance imaging (MRI) contrast agents (CAs), Gd/Y hydroxide nanosheets were synthesized by a simple exfoliation method from layer compounds using sodium polyacrylate (PAA) as a dispersant and stabilizer. Transmission electron microscopy (TEM) and atomic force microscopy (AFM) results revealed the excellent performance of monolayer nanosheets with thicknesses of up to 1.5 nm. The MRI results of the T_1_ and T_2_ relaxation times showed that all of the Gd/Y hydroxide nanosheets have high longitudinal and transverse relaxivities (r_1_ and r_2_). In particular, the 10% Gd-LRH nanosheets exhibited excellent MRI performance (r_1_ = 103 mM^−1^ s^−1^, r_2_ = 372 mM^−1^ s^−1^), which is rarely reported. Based on the relationship between the structure of 10% Gd-LRH nanosheets and their MRI performances, and the highly efficient MRI of spaced Gd atoms in the nanosheets, a special model to explain the outstanding MRI performance of the 10% Gd-LRH nanosheets is suggested. The cytotoxicity assessment of the 10% Gd-LRH nanosheets, evaluated by CCK-8 assays on HeLa cells, indicated no significant cytotoxicity. This study presents a significant advancement in 2D nanomaterial MRI CA research, with Gd-doped nanosheets positioned as highly efficient T_1_/T_2_ MRI CA candidates.

## 1. Introduction

As a solution to Gd leakage from Gd (III) chelates and nephrogenic systemic fibrosis (NSF) [1,2], scientists have focused on synthesizing harmless contrast agents (CAs) with high relaxivity and excellent stability at a low dose. Several kinds of nanostructured magnetic resonance imaging (MRI) CAs have been developed. Bu et al. confirmed that the Gd^3+^ ions bound on the surface of nanoparticles (NPs) play a major role in accelerating relaxation, whereas the contribution from inner Gd^3+^ ions could be neglected [3]. To further reduce the consumption of Gd, monolayer Gd nanosheets are the best alternatives to CAs.

Theoretically, Gd compounds can be good dual-modal CAs. Research has shown that CAs with Gd as the magnetic center exhibit excellent T_1_/T_2_ dual-mode imaging possibilities [4]. Recently, most dual-mode T_1_ and T_2_ MRI CAs have been designed as NPs with Gd shells [5,6,7,8]; in this study we propose a special 2D-structured Gd compound with remarkable T_1_ and T_2_ imaging functions.

Layered rare-earth hydroxides (LRHs), as typical and versatile layered materials, have demonstrated potential as MRI CAs [9,10]. Zhou et al. and Yoon et al. synthesized layered gadolinium hydroxychloride (LGdH) suspensions to enhance the contrast in MR, and they achieved r_1_ values of 2.20 mM^−1^ s^−1^ and 1.7 mM^−1^ s^−1^, and r_2_ values of 6.92 mM^−1^ s^−1^ and 31.6 mM^−1^ s^−1^ [11,12]. However, the relaxation effect of the LGdH suspensions was below expectations, suggesting there are still many obstacles to overcome. Due to the relatively high thickness of the suspension, Gd^3+^ is not adequately exposed, which impedes the performance of the suspension [13]. In addition, most LGdH layers are exfoliated and dispersed in an organic solvent system; thus, because of their poor hydrophilicity, they cannot efficiently influence water protons. The water dispersibility of these nanosheets significantly impacts MRI performance.

In this study, sodium polyacrylate (PAA) was employed because of its abundant hydrophilic and hydrophobic groups, in order to improve the stability and dispersion of nanosheets during the exfoliation process, and Gd^3+^ ion-based monolayer nanosheets were explored as dual-modal CAs [14]. Furthermore, to reduce the consumption of Gd, the MRI performance of nanosheets doped with different Gd^3+^ contents was investigated.

High-quality CAs of hydrotalcite-like-containing rare-earth Gd materials have rarely been reported, primarily because it is difficult to achieve a uniform and stable dispersion of ultrathin nanosheets in an aqueous solution, a challenge which gravely restricts the development of layered materials in CAs [15,16]. Thus, we chose PAA as a stabilizer to promote the exfoliation of nanosheets and improve their stability in aqueous solutions.

Here, we report on ultrathin 2D monolayer nanosheets with different Gd/Y ratios that have superhigh T_1_/T_2_ relaxation performance and good biocompatibility. Among the obtained materials, 10% Gd-doped nanosheets exhibited the highest relaxivity (r_1_ = 103.60 mM^−1^ s^−1^, r_2_ = 372.86 mM^−1^ s^−1^). The relationship between the structure of Gd-doped nanosheets and their MRI performance was investigated. Finally, no significant cytotoxicity was found, which lays the foundation for further study.

## 2. Materials and Methods

Y_2_O_3_ and Gd_2_O_3_ were obtained from J&K Chemicals (J&K Scientific, Beijing, China). PAA (Mw ~5100, RG) was purchased from Sinopharm Chemicals (Sigma-Aldrich, St. Louis, MO, USA). All the chemicals and reagents in this study were of analytical grade and were used without further purification.

### 2.1. Synthesis of the Gd/Y Hydroxide Nanosheets with Different Gd/Ln Ratios

The Gd/Y-LRHs were synthesized using a similar method [17], but with different Gd/Ln ratios: 0.05, 0.1, 0.3, and 1. The exfoliation of Gd-doped yttrium hydroxide nanosheets was conducted as follows: the nanosheet colloidal sol was produced by adding the above-mentioned solid (0.1 g) and 0.1 g of PAA to 200 mL of distilled water after ultrasonic treatment for 10 min using ultrasonic cell-break under ice water bath. After standing overnight, the clarified nanosheet sol was obtained after removing precipitation.

### 2.2. Measurements

The layered structure of LGdHs was validated and characterized by X-ray diffraction (XRD; D8 Advance, Bruker AXS Co., Ltd., Karlsruhe, Germany). The morphologies of the YGdEu–LRH nanosheets were verified by TEM (JEM2100, Hitachi, Tokyo, Japan) and AFM (Multimode 8, Bruker, Madison, WI, USA). The Zeta potential of the nanosheet colloidal solution and the dynamic light scattering measurements of nanosheets were confirmed by Zetasirer nano ZS (Zeta; Zetasirer nano ZS, Malvern, UK). The hysteresis loops (at 300 K) were recorded by a Quantum Design PPMS-9 system (PPMS-9, Quantum Design, Beijing, China). The T_1_ and T_2_ relaxation times of the Gd/Y-LRHs nanosheets were measured at 298 K on a 3.0 T clinical MRI scanner (3T Siemens, Siemens, Munich, Germany). T_1_ was measured using an inversion recovery (IR) pulse sequence, and T_2_ was measured using a Carr–Purcell–Meiboom–Gill (CPMG) spin-echo (SE) pulse sequence. The IR pulse sequence was adopted with the following parameters: _TR_/_TE_ = 7000 ms/11 ms; TI = 24, 100, 200, 400, 600, 900, 1200, 2000, 3000, and 5000 ms. The SE pulse sequence was adopted with the following parameters: _TR_ = 3000 ms, _TE_ = 15.2, 30.4, 45.6, 60.8, 76, 91.2, 106.4, 136.8, 152, and 167.2 ms. The cytotoxicity of the 10% Gd-LRH nanosheets was evaluated by CCK-8 assays. Six different concentrations (0, 0.0125, 0.05, 0.1, 0.2, and 0.4 mM) of the 10% Gd-LRH nanosheets were incubated with HeLa cells for 24 h.

## 3. Results

### 3.1. Characterization of Gd-Nanosheets

Sodium polyacrylate, an anionic surfactant, was introduced into the exfoliating and anti-aggregation process with nanosheets, which tremendously improved the dispersibility and stability. Figure 1a shows the stable, clear, and transparent sol of nanosheets with different Gd/Y ratios, and the typical Tyndall scatter effect, which suggests that the nanosheets were exfoliated from LRHs and steadily dispersed in the aqueous solution. The hysteresis curves indicated that 100% Gd-NS, 30% Gd-NS, 10% Gd-NS, and 5% Gd-NS were paramagnetic at 300 K (Figure 2a). In Figure 2b, the magnetic susceptibility of gadolinium increased with the linear increase in the content (R^2^ = 0.99067), but the growth trend of relaxation properties appeared to be different.

The transparent colloids can be stable for more than three months without aggregation. Some researchers have reported that LRH nanosheets have a positive charge owing to their RH layers. However, our zeta potential (Figure 1b) analysis results revealed that the Gd/Y nanosheets were negatively charged (ca. −35 mV). Because numerous anionic PAA with carboxylic groups were absorbed, the surface charge of Gd/Y nanosheets changed from positive to negative. Consequently, the dispersion and stability of the nanosheets were greatly improved, and monolayer nanosheets could be prepared. The morphology of the as-prepared Gd/Y nanosheets was examined by transmission electron microscopy (TEM) and atomic force microscopy (AFM).

A typical TEM image of the Gd/Y nanosheets in Figure 1d displays a very faint contrast, which indicates that Gd/Y hydroxide layers can be sufficiently exfoliated by sonication and transformed into very thin nanosheets with relatively uniform shapes and sizes. NPs, with sizes ranging from 50 to 300 nm, exhibit good enhanced permeability and retention (EPR) effects; consequently, they can be retained in the tumor for extended periods [18]. The particle size analysis revealed that the Gd/Y nanosheet sizes were around 100 nm in Figure 1c. Through the use of nanosheets with sizes of ~100 nm, it is possible to achieve passive-targeted contrast imaging of tumors [19]. The 2D nanostructures give nanosheets with larger surface areas than NPs with the same volume: the thinner the 2D material, the larger its specific surface area.

AFM images provide a more intuitive cognition of the morphology, particularly regarding the thickness of the nanosheets. A drop of diluted LGdH nanosheet colloidal sol was cast onto a green mica substrate, and the exfoliation was examined by AFM. Figure 1e,f shows the AFM height profile of ultrathin sheets with lateral dimensions of up to 100 nm, and the nanosheets were fairly flat and smooth with an average thickness of ~1.5 nm. The nanosheets were slightly thicker than the monolayer (0.93 nm) [20], which can be attributed to the surface capping ligand of PAA on the nanosheets in the form of brush [7,15,16]. To prevent the leakage of Gd, the PAA immediately protects the nanosheets from aggregation when they are freshly peeled and chelated by Gd^3+^ through carboxyl groups. Simultaneously, PAA creates an extremely water-friendly environment and increases the biocompatibility of the nanosheets to affect more water molecules. The single-layer nanosheet structure not only embeds Gd^3+^ in 2D materials to reduce the risk of Gd leakage, but it also completely exposes every Gd^3+^ ion to the aqueous solution. Therefore, each Gd^3+^ ion in single-layer nanosheets is twice as likely to perturb the magnetic properties of surrounding protons in an aqueous solution compared to the surface-bound Gd^3+^ions on NPs [21].

### 3.2. r_1_ and r_2_ Relaxivity Measurement and MRI in vitro

Enhancing the rotational correlation time (τR) for T_1_ CAs, based on the classical Solomon–Bloembergen–Morgan (SBM) theory, is a common strategy for enhancing relaxivity [22]. Compared to some simple Gd complexes, NPs can prolong τR because of their large sizes. However, when the size of an NP decreases, the enhancement effect is weakened. Two-dimensional nanosheets do not face this dilemma. The decreasing layer thickness not only increases the specific surface area, but it also results in a higher τR value than that for the same volume of NPs [23]. This ultrathin 2D structure makes it a promising MRI CA.

As expected, the MRI performance of nanosheets with different Gd/Y ratios was examined in detail. The samples of 100% Gd-NS, 30% Gd-NS, 10% Gd-NS, and 5% Gd-NS represent Gd3+ molar percentage ratios of 100, 30, 10, and 5 in nanosheets. The T_1_ and T_2_ relaxation times of the Gd/Y nanosheets with different concentrations were measured using a 3T MR scanner, and the corresponding r_1_ and r_2_ values were determined from the slopes of the 1/T_1_ and 1/T_2_ versus the Gd^3+^ concentration (Figure 3a,b). As shown in Table 1, all samples of this study exhibited better performances as T_1_/T_2_ dual-modal CAs than many reported NP CAs, and they significantly exceeded the results previously reported for LRH suspensions (r_1_ even greater than 100 mM^−1^ s^−1^). These results can be attributed to the monolayer nanosheets with the thickness of 1.5nm, good hydrophilicity, and high dispersibility in water. The bar charts of r_1_ and r_2_ values with different Gd/Y ratios are shown in Figure 4a,b. With an increase in the Gd/Ln (lanthanide) molar ratio, r_1_ and r_2_ increased from 44 to 103 mM^−1^ s^−1^ and from 96 to 372 mM^−1^ s^−1^, respectively, and then decreased to 55 mM^−1^ s^−1^ and 90 mM^−1^ s^−1^, respectively, while the variation trend of r_2_ was the same as that of r_1_. Moreover, the 10% Gd-LRH nanosheets, instead of the pure Gd nanosheets, exhibited the best T_1_/T_2_ relaxivities.

The T_1_- and T_2_-weighted images of all the samples at 3T were acquired, as shown in Figure 3c,d. The signal intensity of the T_1_- or T_2_-weighted images gradually increased with the concentration of all the nanosheets, and the 10% Gd-LRH nanosheets showed stronger enhancement than others at an equivalent concentration. These results suggest that the 10% Gd-LRH nanosheets are the best candidates for application as T_1_/T_2_ CAs.

### 3.3. Cell Viability

The cytotoxicity of the LRH nanosheets was evaluated by CCK-8 assays on HeLa cells. Six different concentrations (0, 0.0125, 0.05, 0.1, 0.2, and 0.4 mM) of the 10% Gd-LRH nanosheet sol were incubated with HeLa cells for 24 h. We found that these nanosheet sols exhibited low cytotoxicity. As shown in Figure 5, the activity of the control cells was 100%. When the cells were incubated with different concentrations of the 10% Gd-LRH nanosheet sol, the activity of the cells was more than 90%, which indicates that they had no significant cytotoxicity toward the cells.

## 4. Discussion

In a classical model, the influence of the interaction between water protons and magnetic centers on relaxivity can be considered as inner-sphere and outer-sphere relaxation [25]. Gd^3+^ ions were firmly embedded in these ultrathin nanosheets, which exhibited relatively slow molecular tumbling, and thus high τ_R_ values. Moreover, the special single-layer structure, which can afford naked Gd, has the biggest potential to perturb the magnetic properties of surrounding protons. Meanwhile, LRH nanosheets possess many outer-sphere protons because of aplenty hydroxyls, coordination waters, and strong hydrophilicity. The relaxation is strongly influenced by outer-sphere water molecules. When r_1_ and r_2_ attain maximum values, the Gd/Ln molar ratio is 0.1. Therefore, the 10% Gd-LRH nanosheets were the best T_1_/T_2_ double-weighted CAs, with an r_1_ of 103.6 mM^−1^ s^−1^ and r_2_ of 337 mM^−1^ s^−1^. It is necessary to appropriately elucidate why the special structure of single nanosheets enabled the 10% Gd-LRH nanosheets to exhibit good performance.

Figure 6 shows the structural schematics of the 10% Gd-LRH nanosheets: the structure of the basic metal arrangement is similar to a hexagonal symmetry. There are two types of Ln-coordination spheres surrounded by hydroxyls and H_2_O molecules due to the high and flexible coordination number of Ln: 8-fold dodecahedron polyhedron or 9-fold monocapped square antiprism. Therefore, Ln^3+^ cations were connected by bridged hydroxyl groups, with one coordination water molecule for each Gd^3+^ and two coordination water molecules for each Y^3+^ [26]. We simplified the ultrathin monolayer LRH nanosheet structure to a planar 2D hexagonal grid structure, in which Ln^3+^ cations were uniformly dispersed in the grid points. As the white double arrow in Figure 6a shows, when doped with ~10% Gd, the minimum distance between Gd^3+^ and Gd^3+^ is equivalent to two Y^3+^. Therefore, the white dotted line in Figure 6a describes a basic cell in the 10% Gd-doped single-layer nanosheets. A cell of approximately 1.3 nm includes six adjacent and six sub-adjacent Y^3+^ around each Gd^3+^. Within the scope of a cell, each Gd^3+^ can perturb the relaxation time of not only the protons of the water molecules directly being coordinated with Gd^3+^, but also the protons of the water molecules being coordinated with Y^3+^ and the protons of adjacent water molecules bonded with hydroxyl groups by a hydrogen bond [24]. The large number of bonded water molecules in the cell not only increases the number of protons significantly, but it also provides an optimized water residence time (τ_M_), which is also a common strategy for enhancing relaxivity.

The r_1_ values are composed of the inner-sphere and outer-sphere relaxivities. As shown in Figure 6b, the inner-sphere relaxivity originated from the water ligand directly bound to Gd^3+^. Benefiting from the structure of monolayer nanosheets, the maximum exposure of paramagnetic centers makes them more sensitive to water molecules in the surrounding environment, which is more conducive for the exchange of bound water and bulk water. Usually, water molecule interactions with H-bond acceptor(s) in the chelating ligand occur in the outer sphere of Gd complexes; however, in this case, water molecules bonded to bridged hydroxyl groups and Y^3+^ form the outer-sphere in the cell of the 10% Gd-LRH nanosheets. While there are a few bonded water molecules in the outer sphere of Gd complexes, there are many bonded water molecules within a short distance (1.3 nm) of the cell in the 10% Gd-LRH nanosheet CAs [27]. The outer-sphere relaxation also consists of water ligands, which are directly bound to Y^3+^, and the water molecules form hydrogen bonds with the bridged hydroxyl oxygens, which are directly bound to Gd^3+^. Therefore, a significant amount of water molecules is present in the outer sphere, which contributes significantly to the relaxation properties of the 10% Gd-LRH nanosheets and exhibits the highest utilization of each Gd^3+^ ion and an excellent MRI performance.

For the 10% Gd-LRH monolayer nanosheets, the spin-canting effect cannot be neglected. Meanwhile, the Gd^3+^ ions scattered on the nanosheets further enhance the microscopic magnetic field inhomogeneity, dramatically increase the surface spin disorder, and greatly shorten the T_2_ relaxation time of surrounding water molecules. In this configuration, the T_2_ relaxivity of the 10% Gd-LRH nanosheets is close to that of superparamagnetic iron oxide (SPIO) [28,29]. In addition, PAA has a strong hydrogen-bonding capacity for water molecules, which has a significant, positive impact on both T_1_ and T_2_ relaxivities.

## 5. Conclusions

In summary, Gd/Y hydroxide monolayer nanosheet sols with good dispersion and stability were obtained through the use of PAA as a stabilizer and dispersant,. Because a special monolayer structure can promote slow tumbling and provide numerous sites for aqua ligands from every Gd^3+^ ion, the as-obtained nanosheets, as highly efficient MRI CAs, exhibited excellent dual-modal performance. Particularly, the CAs of the 10% Gd-LRH nanosheets, rather than those of pure Gd nanosheets, showed the highest relaxation properties(r_1_ = 103.60 mM^−1^ s^−1^, r_2_ = 372.86 mM^−1^ s^−1^). According to the structure of the 10% Gd-LRH nanosheets, when two Gd^3+^ ions in nanosheets are exactly two Y^3+^ ions apart, each Gd^3+^ ion in a relatively short distance (1.3 nm) can affect the maximum number of water molecules, and an excellent T_1_/T_2_ MRI performance can be achieved. In addition, the cytotoxicity results revealed no significant cytotoxicity of the Gd-doped nanosheets. The above results validate the potential of Gd-doped nanosheets as safe and highly efficient T_1_/T_2_ double-weighted MRI CAs.

## Figures and Tables

**Figure 1 nanomaterials-11-00017-f001:**
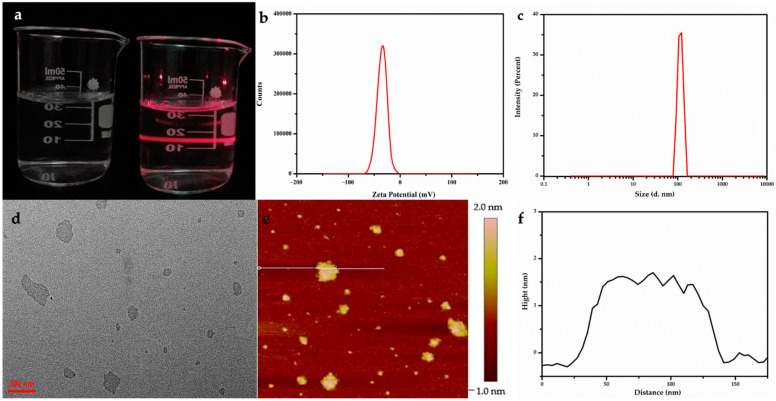
Stable, clear, and transparent sol of Gd/Y nanosheets obtained by ultrasound display typical Tyndall scatter effects (**a**). Zeta potential (**b**) and particle size (**c**) of nanosheet colloidal solution are confirmed to be −35 mV and ~100 nm. TEM (**d**) and AFM (**e**,**f**) characterizations of the Gd/Y nanosheets. Black line in (**f**) represents the height profile of the section in (**e**) labeled with the white line.

**Figure 2 nanomaterials-11-00017-f002:**
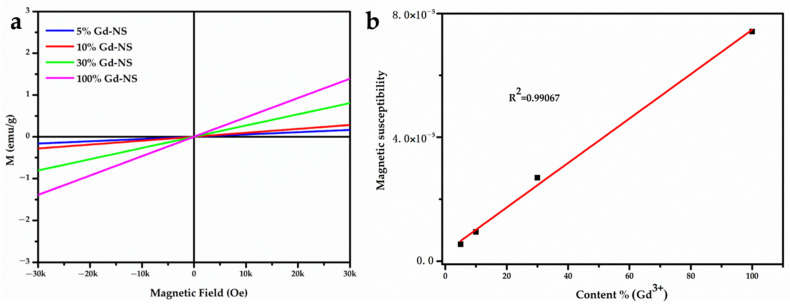
Magnetic hysteresis (M–H) loops of Gd/Y nanosheets at 300 K (**a**). The linear relationship between magnetic susceptibility and gadolinium content (**b**).

**Figure 3 nanomaterials-11-00017-f003:**
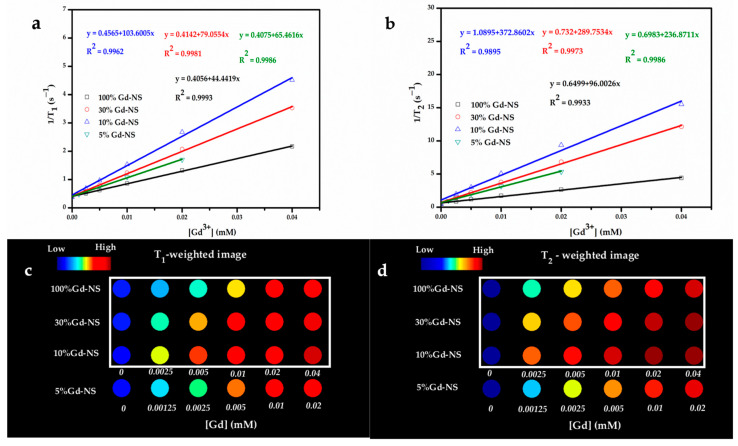
T_1_ relaxivity plot (**a**), T_2_ relaxivity plot (**b**), T_1_-weighted phantom images (**c**), and T_2_-weighted phantom images (**d**) of the samples as a function of the Gd^3+^ concentration (at 3T). The values of the color-coded scale with no unit in (**b**) refer to the gray level of each pixel of images, ranging from 0 (black) to 255 (white); these could be regarded as the signal intensity, and a high value corresponds to a high signal intensity and a bright image.

**Figure 4 nanomaterials-11-00017-f004:**
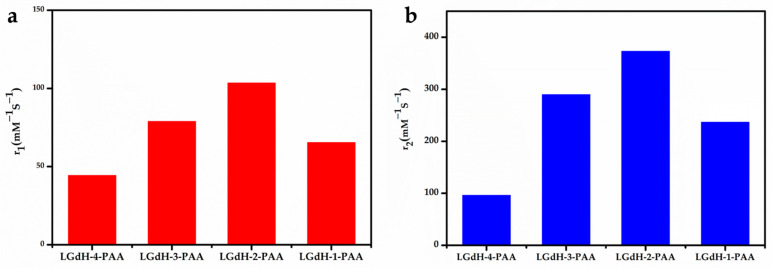
The r_1_ (**a**) and r_2_ (**b**) relaxivities (mM^−1^s^−1^) of a series of Gd^3+^-doped LRH-nanosheet sol.

**Figure 5 nanomaterials-11-00017-f005:**
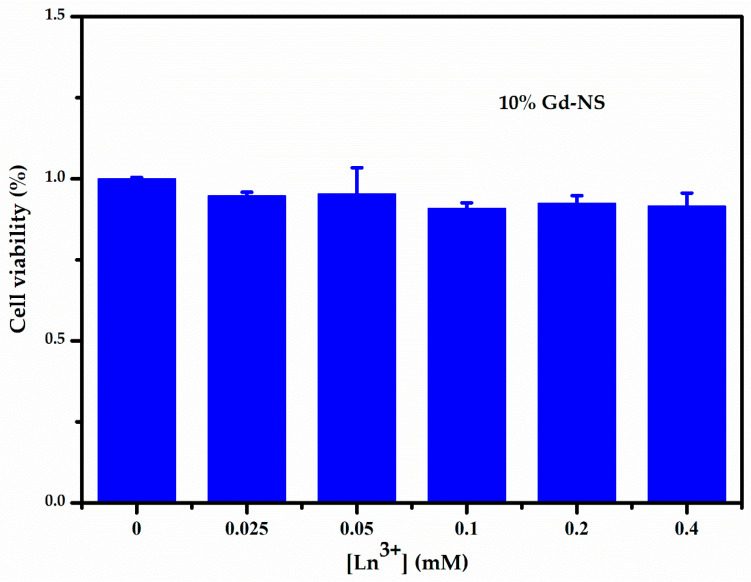
CCK-8 assay result of the cell viability of HeLa induced by 10% Gd-LRH-NS after 24 h.

**Figure 6 nanomaterials-11-00017-f006:**
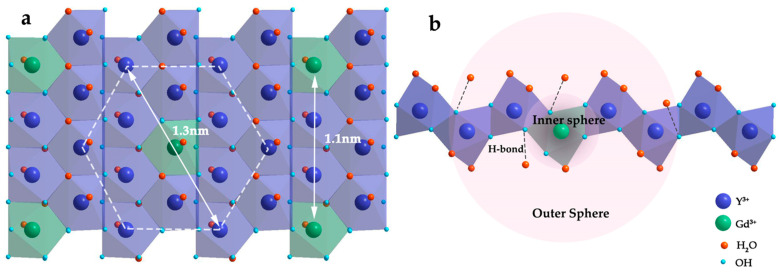
Structural schematics of the 10% Gd-LRH nanosheets (**a**) and cross section structure of basic cell in the 10% Gd-LRH nanosheets (**b**).

**Table 1 nanomaterials-11-00017-t001:** Summary of magnetite nanoparticles for r_1_ and r_2_ relaxivity.

Reference	Simple Code	Field	Magnetic Materials	r_1_ (mM^−1^ s^−1^)	r_2_ (mM^−1^ s^−1^)
This study	100% Gd-NS 30% Gd-NS 10% Gd-NS 5% Gd-NS	3.0 T	monolayers LGdH nanosheets	44.44 79.05 103.60 65.46	96.00 289.75 372.86 236.87
Qiu et al. [5]	Fe_3_O_4_/Gd_2_O_3_ nanocubes	1.5 T	core–shell iron and gadolinium oxide	45.24	186.51
Wu and Chen et al. [15]	ES-GON-PAA,< 2 nm	1.5 T	Gd_2_O_3_	72.10	72.16
Chen and Wu et al. [7]	FeGd-HN_3_-RGD2, 8.5 nm	1.5 T	core–shell Fe_3_O_4_ and Gd_2_O_3_,	70.0	139.2
Gao et al. [24]	Gd_2_O_3_ nanoplates	0.5 T	Gd_2_O_3_	14.5	/
Chuburu et al. [14]	GdDOTA ⊂ NPs	1.5 T	Hydrogels Incorporating Gd chelates	72.3	177.5
Zhou et al. [6]	GdIO	0.5 T	gadolinium hybrid iron oxide	70.10	173.55
Wang, Sun and Yan et al. [16]	PAA-capped GdOF NPs, 2.1 nm	0.5 T	NaGdF_4_	75	81
Lee et al. [12]	LGdH-FS-PEGP	3.0 T	layered gadolinium hydroxide	1.72	31.56
Byeon et al. [11]	LGdH	3.0 T	[Gd_2_(OH)(H_2_0)_x_]CI	2.20	6.92
Yan et al. [23]	PAA-capped GdF_3_ Nanoplates	0.5 T	GdF_3_	15.8	19.8

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
