# Peer review of "Gd/Y Hydroxide Nanosheets as Highly Efficient T1/T2 MRI Contrast Agents"

_nanomaterials, 2020, doi:10.3390/nano11010017_

Round 1

Reviewer 1 Report

I do not see much novelty and/or even minor impact in this research.

It is surprising that no structures are shown.

The MW of the PAA is not declared.

It is not clear how the exfoliation methods work; the authors do not explain which kind of grafting took place on the surface. What does the PAA do? That might be proved with FT-IR before and after grafting.

How much PAA has been inserted? The characterization is missing since I do not see DSC and TGA that would help, in particular TGA, to understand the amount of PAA deposited.

The cytotoxicity experiments are very common and at that low concentration hardly I would expect tox in vitro. Why just 12 hours have been tested? For a complete work at least 24 and 48 hours should be evaluated and with a couple of cell line as well. The drop to 90% viability is not a proof of “biocompatibility” due to the 0.4 mM conc.

Author Response

Point 1: I do not see much novelty and/or even minor impact in this research.

Response 1: I think your views on the originality of this article are open to question, and I will elaborate on it for you. In recent years, Bu et al. confirmed that the Gd3+ ions bound on the surface of nanoparticles (NPs) play a major role in accelerating relaxation, whereas the contribution from inner Gd3+ ions could be neglected. Due to the demand of Gd-based CAs with high performance and low toxicity, it is urgent to further study the relationship between Gd3+ and relaxivity. The most important innovation of this paper is the discovery that monolayer gadolinium doped nanosheets have higher relaxivity than pure gadolinium, which introduces the concept of “doping-Gd”(only 10%) to achieve much higher relaxation efficiency. This is not only a significant reduction in gadolinium consumption resulting in reduced toxicity, but also a new exploration and improvement of the relaxation mechanism in CAs.

Point 2: It is surprising that no structures are shown.

Response 2:The morphology of the as-prepared Gd/Y nanosheets was examined by transmission electron microscopy (TEM) and atomic force microscopy (AFM). Sasaki et al. and Zhang of our lab had studied the structure of such kind of nanosheets in detail,( J. Mater.chem.2011Mar. 21.19. 6903-6908.doi: 10.1039/c1jm00048a.; Dalton Trans., 2012,41, 1854-1861.doi:10.1039/c1dt11332a.; RSC Adv. 2014Jan, 4, 17648-17652. doi: 10.1039/c4ra01881h; Adv Mat Res. 2010Dec, 177, 269-271. doi:10.4028/www.scientific.net/AMR.177.269) , Based on their results,we discussed the relationship between the relaxation properties and the special structure of nanosheets in Figure 6 in this manuscript.

Point 3: The MW of the PAA is not declared. It is not clear how the exfoliation methods work; the authors do not explain which kind of grafting took place on the surface. What does the PAA do? That might be proved with FT-IR before and after grafting. How much PAA has been inserted? The characterization is missing since I do not see DSC and TGA that would help, in particular TGA, to understand the amount of PAA deposited.

Response 3: The MW of the PAA is ~ 5100 and the purity is RG. We have added this valuein the revised manuscript(line 87). The precursors of benzoic acid intercalation compounds were prepared according to the methods reported by our lab, and the nanosheets sols were obtained by ultrasonic. Meanwhile, ultrasonic exfoliation is a mature and commonly used method for the treatment of layered materials to 2D-nanosheets.(Adv Mater.2020 Jan;32.3.e1806461. doi: 10.1002/adma.201806461.) We need to make it clear that PAA, as a dispersant and stabilizer rather than an intercalator in this manuscripts, can immediately protect the nanosheets from aggregation when the nanosheets are freshly peeled and chelated by Gd3+ through carboxyl groups.The –COOH from PAA can be chelated with Gd3+ and Y3+ and capped on the nanosheets surface in the form of brush to lead the formation of a hydrophilic surface (Adv Mater.2018Auc,e1803163.doi: 10.1002/adma.201803163.; ACS nano 2017Apr, 11,4, 3642-3650. doi:10.1021/acsnano.6b07959.; Small. 2019Oct, 1903422,15, 41, doi: 10.1002/smll.201903422.) Our Zeta potential analysis results revealed that the Gd/Y nanosheets were negatively charged (ca. −35 mV). Since numerous anionic PAA with carboxylic groups were capping, the surface charge of Gd/Y nanosheets changed from positive to negative, and significantly improves the hydrophilicity of the nanosheet. In the sol system, it is difficult to study PAA capped on nanosheets alone by TGA, and the carboxyl groups that coordinate with Ln3+ on nanosheets and free carboxyl groups cannot be distinguished by infrared spectroscopy.

[See Line 87 and 150-151 in the revised manuscript].

Point 3: The cytotoxicity experiments are very common and at that low concentration hardly I would expect tox in vitro. Why just 12 hours have been tested? For a complete work at least 24 and 48 hours should be evaluated and with a couple of cell line as well.

Response 3: Thanks for the referee’s suggestion, our experiment was not well considered. We have revised line 113, 192 and 217-219 in the revised manuscript . Below we replaced the data of cytotoxicity after 24 h incubation. As shown in Figure 5, the activity of the control cells was 100%. When the cells were incubated with different concentrations of the 10% Gd-LRH nanosheet sol, the activity of the cells was more than 90%, indicating no significant cytotoxicity.

Figure 5. CCK-8 assay result of the cell viability of HeLa induced by 10% Gd-LRH-NS after 24 h

[See Line 113, 192, 217-219 and Figure 5 in the revised manuscript].

Point 4: The drop to 90% viability is not a proof of “biocompatibility” due to the 0.4 mM conc.

Response 4: The 0.4 mM conc was reasonable for cytotoxicity in Gd-based MRI CAs. Table 1 shows some concentrations of contrast agents for cytotoxicity. We have changed the conclusion of “biocompatibility” in line 24-25, 83-85, 195-197 and 291-295 of the revised manuscript.

Journal name

DOI

CAs

c(Ln)

Nanotechnology

10.1088/0957-4484/26/36/365102

Gd(III) hydroxypyridinone complexes

25μM

Small

10.1002/smll.201903422

ES-GON5-PAA@RGD2

200 × 106 M

Adv Mater

10.1002/adma.201803163

FeGd-HN3- RGD2

490 × 106 M

[See Line 24-25, 83-85, 195-197 and 291-295 in the revised manuscript].

Reviewer 2 Report

This manuscript reports synthesis and characterizations of Gd/Y hydroxide nanosheets. The results indicated 1.5 nm thickness nanosheets and showed high relaxivity values. In addition, good biocompatibility was observed.

  • What is the role of PAA ? It would be explained in the text.
  • PAA can be grafted on the nanosheet surface. Therefore, low cellular cytotoxicities may be due to this.
  • Could you explain the possibility of using the nanosheet in vivo ?

After authors explain the above questions, I suggest the publication.

Author Response

Point 1: What is the role of PAA? It would be explained in the text.

Response 1: PAA as a dispersant and stabilizer immediately protects the nanosheets from aggregation when they are freshly peeled and chelated by Ln3+ through carboxyl groups, and improve their hydrophilicity in aqueous solutions. The –COOH from PAA can be chelated with Gd3+ and Y3+ and capped on the nanosheets surface in the form of brush to lead the formation of a hydrophilic surface (Adv Mater.2018Auc,e1803163.doi: 10.1002/adma.201803163.; ACS nano 2017Apr, 11,4, 3642-3650.doi10.1021/ascnano.6b07959.; Small. 2019Oct, 1903422,15, 41, doi: 10.1002/smll.201903422.) Our Zeta potential analysis results revealed that the Gd/Y nanosheets were negatively charged (ca. −35 mV). Since numerous anionic PAA with carboxylic groups were capping, the surface charge of Gd/Y nanosheets changed from positive to negative, and significantly improves the hydrophilicity of the nanosheet. In addition, the Gd in nanosheets was very hard to be free due to the stabilization by PAA because the Gd ion can be chelated with -COOH from PAA. This does result in low cellular cytotoxicities. However, low dose caused by high relaxation properties is the root cause of reduced toxicity.

[See Line 150-151 in the revised manuscript].

Point 2: Could you explain the possibility of using the nanosheet in vivo?

Response2: The functional elements in the lattice of 2D nanomaterials and molecules can be intercalated into the interlayer or conjugated on the surface of ones to form specific functions materials. Furthermore, therapeutic materials such as anti-cancer drug, gene or gold nanoparticles for chemo-, gene- or photothermal-therapy, respectively, can be additionally hybridized with the 2D nanomaterials. Therefore, 2D nanomaterials can significantly provide advanced information, such as targeting guidance by biodistribution or suitable dosage of drug, for treatment of diseases. (Adv Mater. 2014Nov. 26, 41, 7019.dio:  10.1002/adma.201402572.; Angew. Chem. Int. Ed. 2020Feb.59, 5890 – 5900.doi:10.1002/ange.201911477.; Bull Chem Soc Jpn. 2020Jan. 93, 1, 1-12.doi: 10.1246/bcsj.20190270;Adv. Funct. Mater. 2014Jul. 24, 28, 4386-4396.doi: 10.1002/adfm.201400221.)

Reviewer 3 Report

Dear authors,

thank you for your interesting work on Gd-bearing nanoplatelets, showing a huge effect of relatively low Gd superficial density effect.

Some technical are needed for better understanding:

  • what are the purity and the molecular weight of the PAA you used ? what was the initial concentration? Have you looked for eventual PAA degradation during ultrasonic irradiation (depending of the emission power)?
  • you mentioned some residual compounds during the exfoliation procedure: what is the particle formation ratio? What is the final concentration or your suspensions (please precise the way you determined this concentration) ?
  • what is the amount of free Gd ions in solution ?
  • what is the amount of PAA-bound Gd in solution ? Is it dependent on the amount of PAA molecules you add ? (ie is the PAA molecules able to remove Gd ions from the ytrium platelets ?)
  • what is the superficial density of PAA molecules at the surface of your nanoparticles ? is the coating a dense brush or isolated adsorbed molecules? (this influence the recognition by macrophage system after systemic injection and the EPR effect)
  • you performed one in vitro toxicological test. This is not sufficient to say that these suspensions are biocompatible, they need much more to affirm a such assumption. Please modulate your conclusion.
  • PAA coating of nanoparticles for intravenous injections are known to be poorly tolerated in vivo (Biomaterials. 2019 Nov;222:119442. doi: 10.1016/j.biomaterials.2019.119442. Epub 2019 Aug 22.
    Strategy to prevent cardiac toxicity induced by
    polyacrylic acid decorated iron MRI contrast agent
    and investigation of its mechanism). For a concept model, as you have done in this work, it is very useful but I fear that because of the poor biodegradability, regulatory authorities would be very reticent to allow such suspensions for commercial uses. Have you ideas about biocompatible coatings more adapted to human in vivo uses?

Please complete all these points in you text,

Sincerely yours  

Author Response

Point 1: What are the purity and the molecular weight of the PAA you used? What was the initial concentration? Have you looked for eventual PAA degradation during ultrasonic irradiation (depending of the emission power)?

Response 1: The concentration of PAA ( Mw ~ 5100, RG) at the Ln3+ concentration of 0.4 mM is about 0.1 g/L, and the degradation not be found due to ultrasonic under ice bath. We have completed line 87 in the revised manuscript.

[See Line 87 in the revised manuscript].

Point 2: You mentioned some residual compounds during the exfoliation procedure: what is the particle formation ratio? What is the final concentration or your suspensions (please precise the way you determined this concentration)?

Response 2: The residual compounds during the exfoliation procedure were about 20%. The concentration was determined by ICP ( IRIS Intrepid II ). The specific test steps are as follows: the standard solution concentration of gadolinium ions prepared first is respectively 5 ppm, 10ppm, 15 ppm. Then use pipette will take 5 ml under test including gadolinium nanosheets sol to 10 ml small beaker, then slowly add 2 ml of concentrated hydrochloric acid solution, the sol thoroughly dissolve to gadolinium ion, wait for after cooling to room temperature, moved to the 10 ml volumetric flask and constant volume with distilled water, the solution of the ICP test directly.

Point 3: What is the amount of free Gd ions in solution?

Response3: Some other researchers have reported that LRH nanosheets have a positive charge due to their RH layers. However, our Zeta potential (Figure 1b) analysis results revealed that the Gd/Y nanosheets were negatively charged (ca. −35 mV). The Gd in nanosheet is very hard to be free due to the stabilization by PAA, and then no free gadolinium ions were found in the filtrate after filtration of the colloid with 0.03 μm polycarbonate membrane by ICP.

Point 3: What is the amount of PAA-bound Gd in solution? Is it dependent on the amount of PAA molecules you add? (ie is the PAA molecules able to remove Gd ions from the ytrium platelets ?)

Response 3: We can not quantify the amount of PAA-bound Gd in nanosheet sol, which is to some extent related to the amount of PAA added. PAA molecules cannot able to remove Gd ions from the nanosheets, because Gd in hydroxide nanosheet is very stable and -COOH have less coordination ability than -OH under neutral and slightly acidic.

Point 4: What is the superficial density of PAA molecules at the surface of your nanoparticles ? Is the coating a dense brush or isolated adsorbed molecules? (this influence the recognition by macrophage system after systemic injection and the EPR effect)

Response 4: In the sol system, it is difficult to quantify the density of PAA on the surface, and the carboxyl groups that coordinate with Ln3+ on nanosheets and free carboxyl groups cannot be distinguished. The –COOH from PAA can be chelated with Gd3+ and Y3+ and capped on the nanosheets surface in the form of brush to lead the formation of a hydrophilic surface .(Adv Mater.2018Auc,e1803163.doi: 10.1002/adma.201803163.; ACS nano 2017Apr, 11,4, 3642-3650. doi:10.1021/acsnano.6b07959.; Small. 2019Oct, 1903422,15, 41, doi: 10.1002/smll.201903422.) We have completed line 148-150 in the revised manuscript. Our Zeta potential analysis results revealed that the Gd/Y nanosheets were negatively charged (ca. −35 mV). Since numerous anionic PAA with carboxylic groups were capping, the surface charge of Gd/Y nanosheets changed from positive to negative, and significantly improves the hydrophilicity of the nanosheet.

[See Line 150-151 in the revised manuscript].

Point 5: You performed one in vitro toxicological test. This is not sufficient to say that these suspensions are biocompatible, they need much more to affirm a such assumption. Please modulate your conclusion.

Response 5: Thank you for your suggestion, and we indeed need more to affirm a such assumption. Therefore, we modify the conclusion of "low cytotoxicity and good biocompatibility" in Cell Viability to "no significant cytotoxicity". We have changed the conclusion of “biocompatibility” in line  24-25, 83-85, 195-197 and 291-295 of the revised manuscript.

 [See Line 2 24-25, 83-85, 195-197 and 291-295 in the revised manuscript].

Point 6: PAA coating of nanoparticles for intravenous injections are known to be poorly tolerated in vivo (Biomaterials. 2019 Nov;222:119442. doi: 10.1016/j.biomaterials.2019.119442. Epub 2019 Aug 22.) Strategy to prevent cardiac toxicity induced by polyacrylic acid decorated iron MRI contrast agent and investigation of its mechanism). For a concept model, as you have done in this work, it is very useful but I fear that because of the poor biodegradability, regulatory authorities would be very reticent to allow such suspensions for commercial uses. Have you ideas about biocompatible coatings more adapted to human in vivo uses?

Response 6: Thank you very much for recommending such a high quality literature about PAA toxicity study. This paper is of great significance for our group to further explore the contrast agents with high and low toxicity. Sodium polyacrylate is generally considered as a food additive, so we did not notice its toxicity. (The maximum dosage prescribed by Japan is 0.2% (1993)) Noteworthy, the 10% Gd-LRH nanosheets of this manuscripts has been showed approximately 25-fold higher r1 values and 90-fold than that of commercial CAs (r1 around 4~5 mM−1 s−1, r2 around 5~6 mM−1 s−1). Low doses due to high relaxation can greatly reduce toxicity. The concentration of PAA at the Ln3+ concentration of 0.4 mM is about 0.1 g/L, which may not be very toxic. In addition, the literature you recommended also provides a good solution to handle this kind of problem by pre-chelation Ca2+ or modifying small organic molecules.

Round 2

Reviewer 1 Report

The manuscript has been indeed improved and I appreciate the extra-experiments. I understand the view point fo the authors conning the claim of novelty "monolayer gadolinium doped nanosheets have higher relaxivity than pure gadolinium". I still considered this limited but the improvements render this manuscript now much more than a routine job and brings indeed some impact to the nanotech community.

Paper can ben now accepted as it is